# HISP: Heterogeneous Image Signal Processor Pipeline Combining Traditional and Deep Learning Algorithms Implemented on FPGA

Jie Chen [1,2,3], Binghao Wang [1,2], Shupei He [1,2], Qijun Xing [1,2], Xing Su [1,2], Wei Liu [1,2,4,*] and Ge Gao [3,5,*]

1  School of Physics and Technology, Wuhan University, Wuhan 430072, China; 2018302020114@whu.edu.cn (J.C.)
2  School of Microelectronics, Wuhan University, Wuhan 430072, China
3  Hubei Luojia Laboratory, Wuhan 430072, China
4  Wuhan Institute of Quantum Technology, Wuhan 430072, China
5  School of Computer Science, Wuhan University, Wuhan 430072, China
*  Correspondence: wliu@whu.edu.cn (W.L.); gaoge@whu.edu.cn (G.G.)

**Abstract:** To tackle the challenges of edge image processing scenarios, we have developed a novel heterogeneous image signal processor (HISP) pipeline combining the advantages of traditional image signal processors and deep learning ISP (DLISP). Through a multi-dimensional image quality assessment (IQA) system integrating deep learning and traditional methods like RankIQA, BRISQUE, and SSIM, various partitioning schemes were compared to explore the highest-quality imaging heterogeneous processing scheme. The UNet-specific deep-learning processing unit (DPU) based on a field programmable gate array (FPGA) provided a $14.67\times$ acceleration ratio for the total network and for deconvolution and max pool, the calculation latency was as low as 2.46 ms and 97.10 ms, achieving an impressive speedup ratio of $46.30\times$ and $36.49\times$ with only 4.04 W power consumption. The HISP consisting of a DPU and the FPGA-implemented traditional image signal processor (ISP) submodules, which scored highly in the image quality assessment system, with a single processing time of 524.93 ms and power consumption of only 8.56 W, provided a low-cost and fully replicable solution for edge image processing in extremely low illumination and high noise environments.

**Keywords:** image signal processor (ISP); deep learning (DL); image quality assessment (IQA); FPGA; hardware implementation

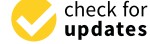

## 1. Introduction

From globally popular smartphones and digital cameras to DSLRs and surveillance cameras, the image signal processor (ISP) is ubiquitous [1]. Since the raw data converted by light signals into digital signals from CMOS or CCD image sensors do not meet the expectations of human eyes and most computer vision recognition algorithms [2], and physical defects such as distortion, bad pixels, and dark current exist in lenses and sensors that require correction, the ISP pipeline has become an indispensable part of image processing. In the past two years, due to the rise of autonomous driving, virtual reality, and drones, the requirements for image acquisition and analysis have increased dramatically, and ISP algorithms have played a cornerstone role in image processing in many of the latest camera applications (such as YOLOv5-tassel [3] in 2022 and HYDRO-3D [4] in 2023). Research on the impact of ISP image quality is also on the rise [5].

An ISP consists of a series of image processing modules connected in a pipeline structure, which enhances image quality by performing various processing tasks such as noise reduction, sharpening, enhancement, and color correction [6]. Because the operations and parameters of each submodule are usually fixed, the traditional ISP is reliable, consistent, and predictable in terms of performance [7]. However, it is difficult for it to handle complex

scenes with different lighting conditions [8], and achieving good imaging results requires a significant amount of tuning work on the parameters and pipeline composition [9]. Although image processing based on traditional algorithms has also made considerable progress in the fields of dehazing, denoising, and so on [10–13], the above problems have not been effectively solved. At the same time, it is clear that the superiority and convenience of image processing through neural networks have sparked more discussions in journals and at conferences.

In recent years, with the development of deep learning, more and more research has attempted to use deep learning methods to improve image quality in a particular dimension or even to simulate the entire traditional ISP pipeline [14]. The emergence of fully convolutional networks (FCN) [15] and UNet [16] in 2015 greatly reduced the amount of data required for training of deep learning neural networks, which originally required thousands of annotated data for training, and created a precedent for the application of neural networks in image segmentation. Since then, algorithms have emerged to solve end-to-end super-resolution (SR) problems, including SRCNN (the first trial to solve SR problems using a CNN structure) [17], VDSR using a ResNet structure [18], and SRGAN using generative adversarial networks [19]. However, the above studies were all deep learning schemes based on a single problem, until the paper "Learning to See in the Dark", published in 2018, used UNet to solve the imaging problem in low-light environments [20]. The model used the RAW Bayer format as input and RGB as output, which was a complete ISP from an imaging perspective and set off a boom in replacing entire ISP pipeline functions through a single deep learning model.

In order to cope with more complex environments and produce better imaging results, the researchers proposed a full end-to-end deep neural model called DeepISP [21] in 2018, a novel pyramid CNN architecture for fine-grained image restoration named PyNet [22], and CycleISP, which achieved state-of-the-art performance on real camera benchmark datasets [23] in 2020. The more recent and interesting studies are on CSANet [24] and PyNET-CA [25], and both of them employ the channel spatial attention method in their networks. A series of image processing units implemented through deep learning, such as DeepISP, PyNet, CycleISP, and CSANet, are typically referred to as deep learning ISP (DLISP). In contrast, DLISP does not have fixed submodules to handle different functions, nor does it require organizing fixed pipelines and manually adjusting parameters. Instead, it often simulates the entire ISP pipeline's processing effects through a single end-to-end neural network [26] (such as a fully convolutional neural network like UNet or WNet).

Due to its ability to extract complex and multidimensional parameters from large amounts of targeted training sets, DLISP has stronger adaptability and superior processing performance in specific scenarios [27]. Moreover, it is easier to optimize and iterate the model without requiring a significant amount of manual intervention. However, DLISP requires significant computing power and storage resources. Most research is focused on improving the quality of the output image but overlooks the high real-time requirements and limited computing resources in which reproducing these effects poses a significant challenge. As a result, although academia and industry have proposed many DLISP models that often outperform traditional ISP pipelines in extreme environments such as low light and high noise, the lack of device computational power means that few DLISP models can be effectively deployed for production and use in real-time scenarios [28]. Especially in edge scenarios, it is impossible to integrate high-performance CPU and GPU clusters to infer neural networks for real-time processing. Therefore, in order to enable deep learning to play a more reasonable role and create more value in practical scenarios, customized hardware acceleration schemes for deep learning edge inference are necessary. There have been attempts to combine DLISP with traditional methods, but no one has been able to explain well what the division of image processing tasks should be, why such the division should be adopted, and how to better integrate the two to efficiently leverage their strengths.

For the current DLISPs, the real obstacle to their standardization and adoption in production is the computational limitations of edge devices, which result in insufficient inference speed, preventing them from meeting the most basic requirement of real-time performance as an ISP. This requires the design of accelerators to speed up neural network inference in edge scenarios. The mainstream choice for accelerating neural network inference in edge scenarios is through the design of specific hardware accelerators such as NVDLA and other NPUs [29]. However, hardware accelerators with general architectures implemented through ASICs not only have a high design difficulty and a long development cycle but also may not be sufficient to meet the real-time requirements in terms of acceleration ratio. Additionally, general-purpose NPUs often require software and may even need to run an operating system on the CPU for scheduling, which increases development difficulty and results in longer memory access time. Some researchers have proposed solutions that optimize hardware structures for specific models, reducing redundant designs and concentrating hardware resources to improve the inference speed of specific operators [30–32]. However, this approach only makes the implementation of the NPU more difficult and reduces its universality and versatility.

This paper proposes a solution to implement a dedicated DPU for a specific network using FPGA, emphasizing the specificity of the accelerator from the design stage, and implementing the hardware network from specific operators to fully leverage the advantages of parallel computing in FPGAs. This achieves an acceleration ratio far higher than that of a general-purpose NPU, meeting the real-time requirements for image processing. Based on this, this paper proposes the concept of heterogeneous ISP (HISP): by dividing different tasks between DLISP and traditional ISP submodules and combining them, the fitting results of the neural network model are coordinated with the adjustments of the fixed pipeline to output the optimal image quality in various complex and extreme scenarios. By implementing the entire HISP pipeline on FPGA, and utilizing the specialized DPU for accelerating the DL algorithm mentioned above, the remaining resources are used to implement the most important ISP submodules, achieving an exciting processing effect with low power consumption and low latency.

In particular, this paper has three main contributions:

- Detailed analysis of the strengths and weaknesses of traditional ISP and DLISP, and proposal of the concept of HISP to combine the two, leveraging their advantages while minimizing their drawbacks.
- Integration of different traditional ISP modules with DLISP to create multiple pipelines, which will be evaluated through multiple dimensions of image quality assessment (IQA). Proposing an HISP allocation plan that divides processing tasks for traditional and deep-learning modules and achieves the optimal balance among processing speed, resource consumption, and development difficulty.
- Implementation of a dedicated DPU for UNet on FPGA, achieving a $14.67\times$ acceleration ratio. In addition, we designed a heterogeneous ISP that combines traditional ISP and DLISP based on the optimal division of labor, all on FPGA, resulting in the best image quality in edge scenarios and costing only 8.56 W power.

The remainder of this paper is structured as follows:

Section 2 explores works more related to the proposed solution, details about the analysis with comparative experiment are given in Section 3, Section 4 shows the detailed implementation of the hardware system, experimental results are in Section 5, while final remarks are in Section 6.

## 2. Related Work

### 2.1. Traditional ISP Principle and Pipeline

The design focus of the heterogeneous ISP (HISP) is to reasonably allocate tasks between traditional processing methods and deep learning methods, allowing each to play to its strengths, minimizing unnecessary resource waste, and avoiding inefficient processing. The premise of exploring the optimal task allocation scheme is to understand

the basic principles of traditional ISP and DLISP, and to clearly understand the advantages and disadvantages of the two through comparative experiments, which all begin with the processing pipeline of traditional ISP.

A typical ISP pipeline consists of a series of interconnected processing submodules that operate at high speed under clock signals of several hundred MHz. The input of an ISP pipeline usually consists of Bayer mosaic format RAW data from a CMOS or CCD sensor output [33]. The image data are continuously transferred from one submodule to the next until all the processing is completed and finally flow out of the pipeline in the form of YUV or RGB. Figure 1 shows a common basic ISP pipeline.

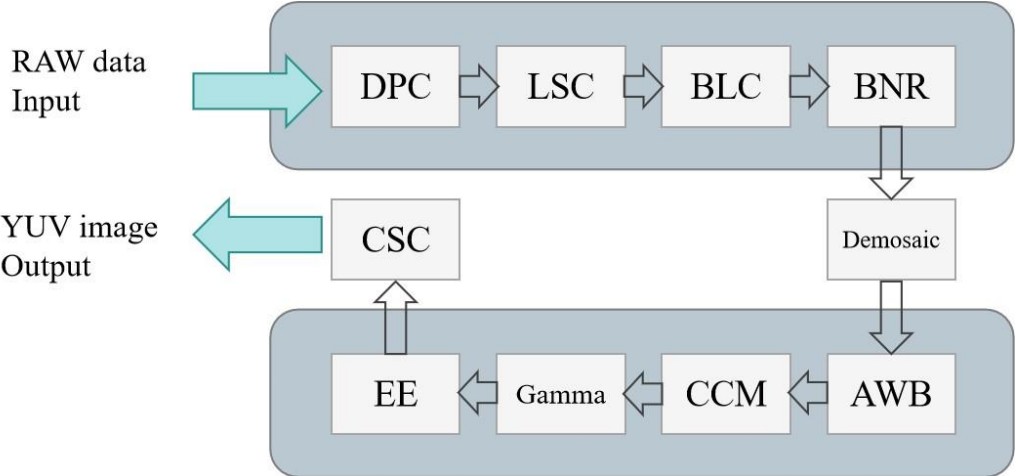

**Figure 1.** Traditional pipeline of image signal processing.

In a camera, the image sensor converts light into electric current, and the value of the current varies with the intensity of the light. This means that the sensor cannot distinguish the wavelength of light. In order to obtain color information, in 1974, engineer Bryce Bayer of Kodak proposed to place a layered color filter array (CFA) in front of the image sensor [34], which avoids the cost and alignment issues of using three different filters.

CFA arranges the color filters in a certain way on the pixels of the sensor, with each pixel filtered by only one color filter, resulting in missing color information. Therefore, the image output by the Bayer array is very unrealistic to the human eye. To restore the complete color image, interpolation algorithms such as bilinear or bicubic interpolation are used to predict and fill in the missing color information based on the arrangement of the Bayer pattern and the information of neighboring pixels [35,36]. This process is usually referred to as demosaicing. The intuitive effect is shown in Figure 2.

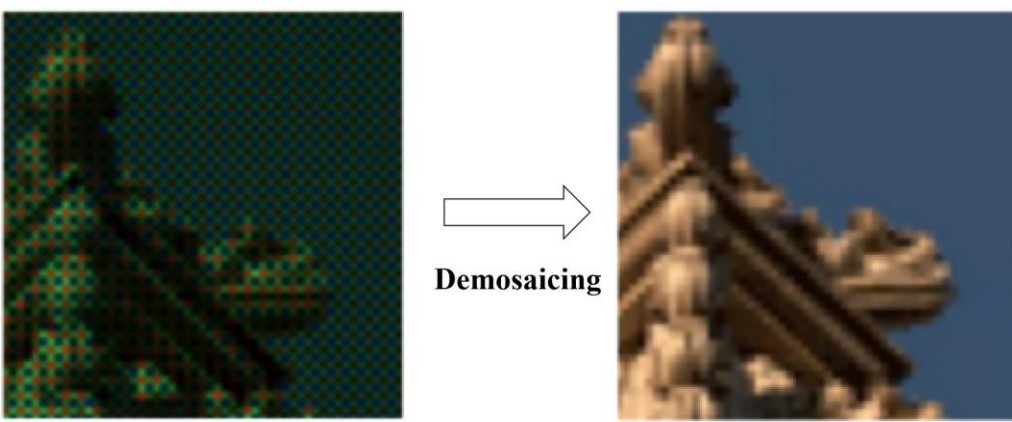

**Figure 2.** Effect of demosaicing submodule in traditional ISP.

Demosaicing is the most important step to make an ISP transform non-intuitive raw data into images that look pleasing to the human eye. Suppose there are currently four pixels with coordinates G1 (x1,y1), G2 (x2,y2), G3 (x3,y3), and G4 (x4,y4): we need to determine the pixel value of a point with coordinates Gx (x,y), so the most common bilinear interpolation algorithm in the demosaicing module principle is as follows [37]:

$$f(T_1) = \frac{x_2 - x}{x_2 - x_1} f(G_1) + \frac{x - x_1}{x_2 - x_1} f(G_2) \tag{1}$$

$$f(T_2) = \frac{x_2 - x}{x_2 - x_1} f(G_3) + \frac{x - x_1}{x_2 - x_1} f(G_4) \tag{2}$$

$$f(G_x) = \frac{y_2 - y}{y_2 - y_1} f(T_1) + \frac{y - y_1}{y_2 - y_1} f(T_2) \tag{3}$$

Substituting the first two equations into the third equation yields:

$$
\begin{aligned}
f(x,y) &= \frac{(x_2-x)(y_2-y)}{(x_2-x_1)(y_2-y_1)} f(G_1) + \frac{(x-x_1)(y_2-y)}{(x_2-x_1)(y_2-y_1)} f(G_2) + \frac{(x_2-x)(y_2-y)}{(x_2-x_1)(y_2-y_1)} f(G_3) + \frac{(x-x_1)(y_2-y)}{(x_2-x_1)(y_2-y_1)} f(G_4) \\
&= \frac{(x_2-x)(y_2-y)}{(x_2-x_1)(y_2-y_1)} (f(G_1) + f(G_3)) + \frac{(x-x_1)(y_2-y)}{(x_2-x_1)(y_2-y_1)} (f(G_2) + f(G_4))
\end{aligned}
\tag{4}
$$

In summary, the whole process is reflected in Figure 3. Demosaicing can be considered a crucial step in image processing for ISP, as it greatly improves both the visual effect for human eyes and the recognition performance of computer vision algorithms [38]. Therefore, correcting the physical defects of the sensor and lens before demosaicing to ensure the quality of Bayer images is also an essential step. This involves bad pixel correction (BPC), black level correction (BLC), lens shading correction (LSC), and Bayer noise removal (BNR) in the Bayer domain.

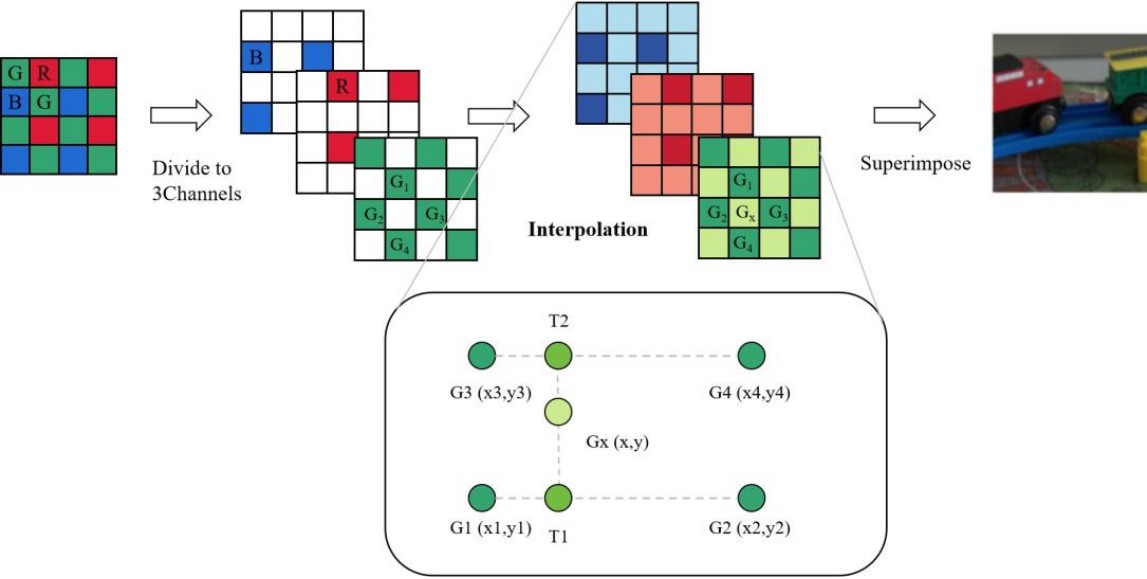

**Figure 3.** Transform Bayer pattern to RGB pattern by bilinear interpolation.

These steps are also uniformly classified into preprocessing in the ISP pipeline. After demosaicing, the image is converted from Bayer's raw data to RGB format through color interpolation. At this point, the complete color image already roughly conforms to the human eye's visual perception. However, the image may still have problems such as being too dark, too bright, color deviation, blur, and noise, which can affect the final output image quality. Therefore, various processing techniques are needed in the RGB domain to improve image quality.

One of the most typical steps is gamma correction, which can enhance the details in the dark areas of the image while adjusting the color [39]. This creates a bright and clear effect for the color image. Gamma correction compensates for the color display differences that exist on different output devices, making the image appear the same on different monitors [40]. It also enhances the dynamic range and detail in the dark areas of the image to better respond to the human eye's sensitivity to dark areas [41].

In addition, the RGB image also needs automatic white balance (AWB), color correction matrix (CCM), denoising, edge enhancement (EE), and other processing techniques. Some ISPs also need to convert the processed RGB domain data into YUV format output through color space conversion (CSC) at the end or in the middle of the pipeline. This can further optimize the data in the YUV domain or be directly output.

Different manufacturers' ISPs may not entirely use the same submodules and organize the pipeline in the same order at this step. However, the overall structure is similar, and different application scenarios and requirements may also result in differences in ISP pipelines. The complex structure also explains why traditional ISPs have poor adaptability and require a lot of manual adjustment of various parameters.

In the traditional ISP pipeline, each of the above processing steps is completed by a single submodule that is connected in series to form a complete pipeline. In addition, there are L3 [42], Burst [43] as shown in Figure 4, and other pipelines, which may have different forms but still generally process through a fixed pipeline.

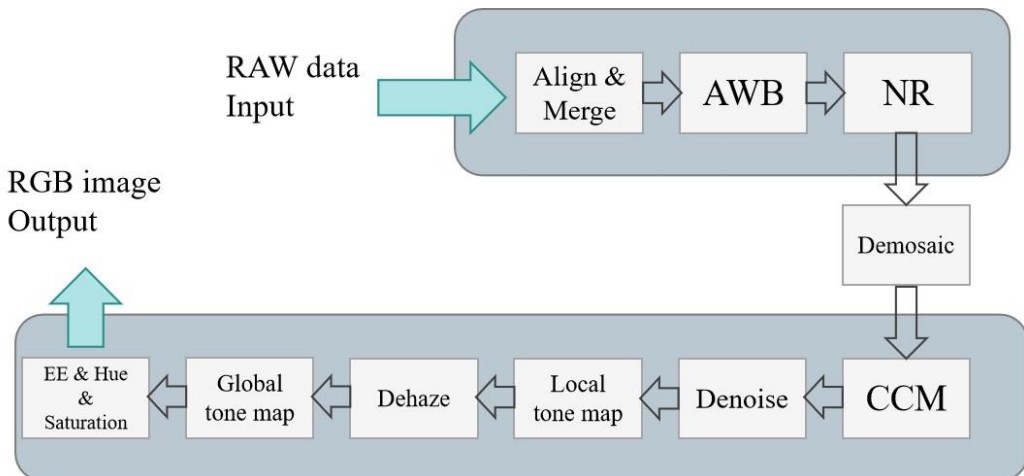

**Figure 4.** Burst image signal processing pipeline.

## *2.2. Deep Learning ISP*

To accurately position the role that deep learning should play in the image processing pipeline, we not only need to understand the principles of ISP but also need a clear understanding of the advantages of deep learning compared to traditional methods.

In earlier years, there were many studies that used deep learning methods for image denoising, dehazing, gamma correction, and brightness enhancement, among other image processing tasks. These studies can be viewed as simulating individual submodules within an ISP through neural networks, thereby achieving improvements in single dimensions of image quality.

In these studies, using deep learning for image denoising has been a particularly popular topic. For instance, DnCNN in 2017 used deep convolutional neural networks for denoising and achieved excellent results, whose feature was using a large number of noisy images to generate training data during training instead of using real images and noise pairs, and MIRNet in 2020 was a multi-scale residual network that could perform denoising and super-resolution simultaneously. Its feature was using depthwise separable convolution to reduce the number of parameters and thus lower computational cost. This method performs well in complex image reconstruction.

There is no doubt that the emergence of a large number of neural networks for denoising and enhancing images is essentially due to the extremely significant advantages of deep learning in these two types of tasks in image processing [44–46].

"Learning to See in the Dark" opened the door to simulate the complete ISP pipeline, and since 2019, there have been numerous experiments and studies using end-to-end neural networks to simulate the entire ISP pipeline [20,47].

For example, DeepISP used a deep neural network to simulate the entire ISP pipeline, including demosaicing, denoising, color correction, and JPEG compression. Its feature was that it could perform end-to-end optimization and achieve better results than traditional methods. This solution achieved state-of-the-art performance in the objective evaluation of the peak signal-to-noise ratio on the subtask of joint denoising and demosaicing. Figure 5 shows the process of a typical DLISP.

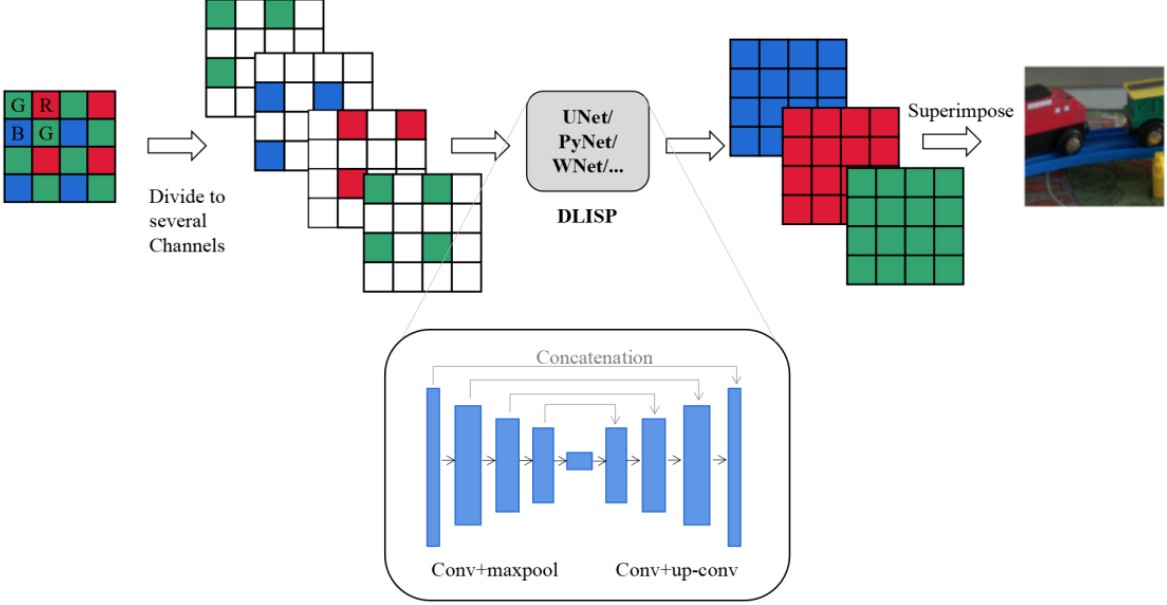

**Figure 5.** Typical DLISP using UNet neural network model. (The RGB tricolor is split into three channels and processed).

In general, the reason why deep learning methods are better than traditional ISP is that they can learn the complex mapping from raw sensor data to the final image more effectively. They can adaptively adjust processing methods to better deal with different types of images and image quality issues. In addition, deep learning methods can use a large amount of training data to learn the statistical properties of natural images, which enables them to generalize better to new images.

Simulating the entire ISP pipeline through deep learning has been proven to produce superior results in extreme environments such as low light, dark areas, and high noise. This further indicates that DLISP is capable of effectively handling both denoising and brightening tasks.

### 2.3. No-Reference Image Quality Assessment Scheme

The deep learning scheme brings a lot of changes visible to the naked eye; however, the intuitive perception of the naked eye is obviously not a scientific criterion for evaluating the quality of an image, so we will build an IQA scheme to accurately quantify the output image quality of traditional ISP, DLISPs, and HISPs.

Image quality assessment (IQA) refers to using a series of mathematical methods and algorithms to measure and evaluate the visual quality of digital images. It can be divided into full-reference (FR), no-reference (NR), and reduced-reference (RR) methods. Over the

past few decades, many traditional and deep learning IQA methods have been developed and studied to improve the accuracy and efficiency of image quality assessment.

In an ISP output image quality evaluation scheme, it is not necessary to pay attention to the content and local statistical characteristics of the picture, such as structure, color, and texture. Instead, it can quantify the macro statistical characteristics of the image, such as clarity, color contrast, edge information, and noise in the face of various content and types of images, as the performance indicator of our image processing flow. Therefore, it is necessary to summarize a reliable no-reference image quality assessment (RRIQA) scheme.

Common traditional NR methods include BRISQUE [48], NIQE [49], PIQE [50], etc. BRISQUE and NIQE are two models based on natural scene statistical features. The BRISQUE method calculates features such as image gradients, contrast, color distribution, and sharpness, while the NIQE method calculates features such as local contrast, gradient statistics, and phase consistency and uses these features to predict image quality. The advantages of these two methods are that they can perform quality assessment without a reference image, but their disadvantages are that they have a certain dependency on the scene and image type. PIQE is a model based on the human visual perception model. It also calculates IQA by calculating features such as contrast, color saturation, and sharpness. However, since this method simulates the image processing in the human visual system, its prediction results have a high correlation with the results of human subjective evaluation, which is more in line with human visual perception.

Since traditional methods are generally highly dependent on the image scene and type [51], their accuracy and robustness in practical applications still need to be further studied and improved. In recent years, with the development of deep learning technology, more and more research has focused on using deep learning methods to solve the problem of image quality assessment [52,53].

The CNN-based deep learning IQA method is mainly based on CNN learning of image feature representation, which can help determine the quality of the image. These methods use a large amount of training data to train the CNN model to predict image quality. Common CNN-based methods include DeepIQA, NIMA, RankIQA, etc.

After research and decision-making, NIMA, proposed by Google's research team, uses a new neural network architecture that adds an attention mechanism when learning the quality characteristics of the image, which can better capture the important information in the image. The core idea of RankIQA is that it is not necessary to make an accurate quality assessment of each image, but instead focuses on comparing quality differences between images, which tends to be more robust in practical applications.

Therefore, in the RRIQA scheme for ISP output image quality with different inputs, NIMA and RankIQA will be combined with BRISQUE, NIQE, and PIQE, three methods of scoring are included as evaluation indicators, and finally, we also add the result of artificial blind scoring (ABS). The purpose is to consider the image quality score under different evaluation systems at the same time and add the score of visual perception to evaluate an image from multiple perspectives as much as possible to ensure the credibility and universality of the results.

The scores of BRISQUE and PIQE are usually two digits within 100, and the range of score changes is in the dozens, while the scores of NIQE, NIMA, and RankIQA are mainly distributed between 1–10, and the score changes reflecting image quality are usually only single digits or even a few tenths. At the same time, ABS conducted a blind evaluation of each picture through 100 questionnaires, and the scoring standard ranged from 1 to 10. Due to the different score ranges, we needed to normalize the different scores first. As a formula used in the initial exploration, based on our experimental experience, we normalized BRISQUE and PIQE by directly dividing by 10 (linear treatment).

The higher the scores of BRISQUE, PIQE, and NIQE, the poorer the image quality, and the higher the scores of NIMA and RankIQA, the higher the image quality. At the same time, the results of different methods had certain differences, so we also needed to synthesize each score. This part of this research has tried to use addition, multiplication, and even

neural network methods, and finally, for the consideration of time cost and convenience, we used simple linear addition. In all the methods, simple additions to normalized scores also performed well.

The final score can be derived from the following formula:

$$\text{Final Score} = \text{NIMA} + \text{RankIQA} + \text{ABS} - (\text{BRISQUE}/10 + \text{PIQE}/10 + \text{NIQE}) \quad (5)$$

For pipelines with the same input, since the score distribution and ranking given by each evaluation system are different, we canceled the calculation of the final score but added the ground truth image as a reference while taking the non-reference IQA. Referring to the IQA system, the parameters evaluated were peak signal-to-noise ratio (PSNR) and structural similarity index measure (SSIM).

## 3. Analysis

### 3.1. How to Allocate the Task?

How to make the traditional ISP submodule cooperate with DLISP and realize the perfect task allocation is actually the most important issue of HISP, which cannot be overemphasized.

Research has shown that among all the ISP functions, demosaicing and gamma correction have the greatest impact on the performance of final computer vision tasks. For example, poor demosaicing can have various negative effects, including zipper artifacts, edge blurring, color errors (false color effect), etc.

Therefore, evaluating the implementation effects of traditional ISP submodules and DLISP on demosaicing and gamma correction can help us make a choice and allocate these two tasks to the best-performing method.

The low-light, high-noise raw data from the SID dataset were selected as input and directly fed into the demosaic and gamma correction submodules of the traditional ISP for processing. We then compared the results with those of the software algorithm and DLISP's processing results as shown in Figure 6.

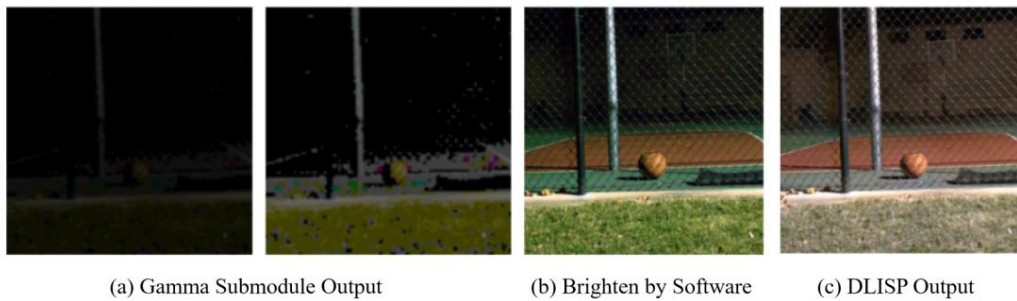

(a) Gamma Submodule Output      (b) Brighten by Software      (c) DLISP Output

**Figure 6.** Image output by gamma submodule, software, and DLISP.

Visibly, the result obtained by directly processing the raw data under extremely low light conditions through gamma correction is extremely poor, and there is a significant difference compared to the results obtained by software algorithms. However, DLISP outputs a brighter and smoother image, and it can be demonstrated that traditional ISPs have weak adaptability in extreme scenarios and often cannot cope with low-light and high-noise environments when demosaicing and gamma correcting.

In theory, DLISP has processing capabilities that traditional ISP cannot match, particularly in extreme scenarios. However, traditional ISP still has its advantages.

To achieve optimal processing results, neural network models have become increasingly complex and voluminous, with continuous growth in model parameters from CNN to UNet to WNet. This means that there are higher hardware requirements for neural network inference. If these algorithms are used on devices with limited computing power, they cannot guarantee basic replicability and robustness, and their processing speed depends

on the size of the device's computing power. They are usually unable to meet real-time requirements in various edge scenarios.

Moreover, DLISP's processing capabilities are not flawless in any scenario. For example, in "Learning to See in the Dark", using the official pre-trained model and setting the brightness enhancement factor to 100 for low-ISO raw images (ISO = 0.1 s) in the official SID dataset, visible information loss, edge blur, noise, and color deviation occur as shown in Figure 7.

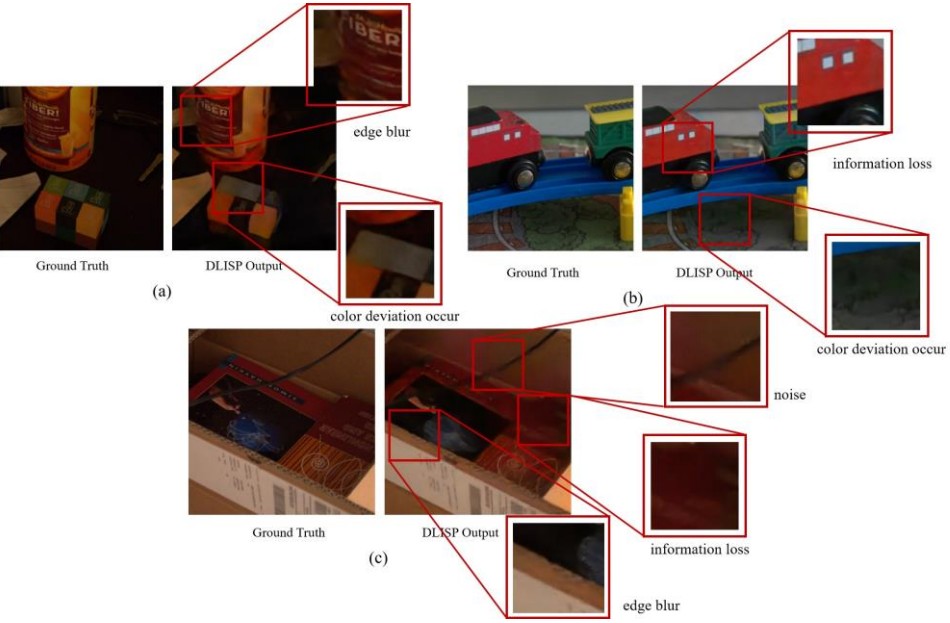

**Figure 7.** Different problems in DLISP's output image. The (**a**–**c**) three sub-figures represent experimental results on three different images, which show that phenomena such as information loss and blurring are common.

Through the previous RRIQA scheme, we were able to accurately quantify the evaluation results. The specific results are shown in Table 1.

**Table 1.** By using a multidimensional IQA system, the differences between the DLISP output image and the ground truth can be obtained.

|  | BRISQUE | PIQE | NIQE | NIMA | RankIQA | ABS | Final Score |
|---|---|---|---|---|---|---|---|
| Gt1 | 12.13 | 24.60 | 3.10 | 5.17 | 4.21 | 7.70 | 10.30 |
| Output1 | 30.66 | 43.55 | 3.77 | 4.66 | 4.05 | 6.30 | 3.82 |
| Gt2 | 24.40 | 16.02 | 3.77 | 4.76 | 4.83 | 8.90 | 10.69 |
| Output2 | 36.80 | 51.58 | 4.31 | 4.62 | 4.39 | 7.30 | 3.16 |
| Gt3 | 22.85 | 12.02 | 4.55 | 4.96 | 4.27 | 8.00 | 9.19 |
| Output3 | 33.27 | 54.02 | 4.65 | 5.19 | 3.73 | 7.10 | 2.64 |

Among them, RankIQA's Pearson linear correlation coefficient (PLCC) = 0.8175 and Spearman rank-order correlation coefficient (SROCC) = 0.7819. Through the final score, we can clearly see the quality difference between the ground truth image and the final output image.

Further increasing the amount of training and the richness of the dataset can make DLISP have a better processing effect.

At the same time, the more similar the input picture scene is to the scene set in the training set, the higher the quality of the output image will be. However, even so, the abovementioned problems of noise, blur, and loss of color information will still occur.

### 3.2. HISP May Work Better

The traditional ISP pipeline can typically achieve several hundred frames per second, and selecting some of its submodules to assist or enhance the processing flow of DLISP does not bring significant delays during implementation. Moreover, the HISP, which combines the two image processing methods, may exhibit superior performance compared to their individual operation.

To preliminarily validate the superiority of the heterogeneous ISP (HISP), we selectively chose several traditional ISP submodules based on various types of defects (such as edge blur, overexposure, and color deviation) in the low IQA score images output by DLISP. These submodules assisted DLISP to form a HISP for processing, and the output of HISP under various scenarios was compared with the processing effects of traditional ISP and DLISP in Figure 8.

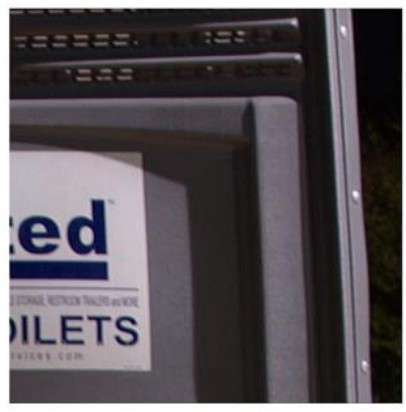
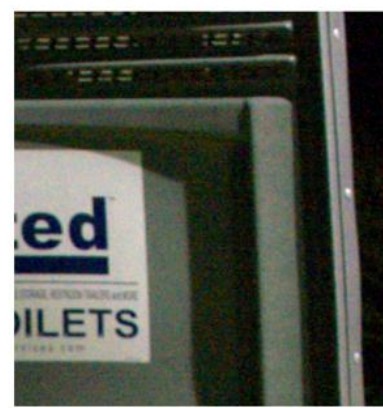
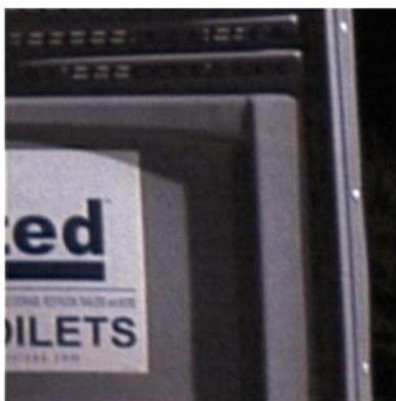

(a) Ground Truth      (b) Traditional ISP Output      (c) DLISP Output

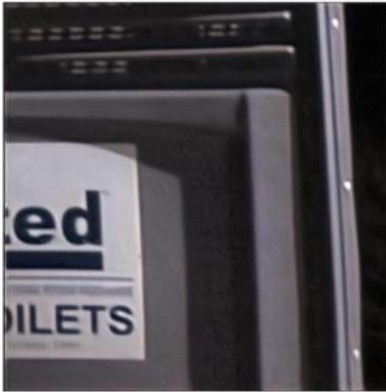
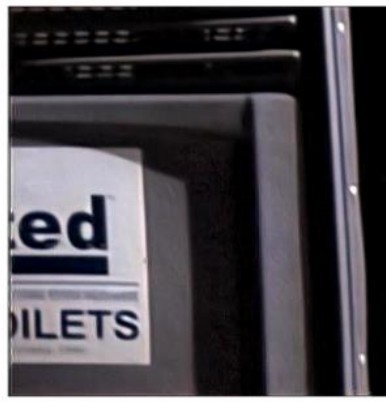
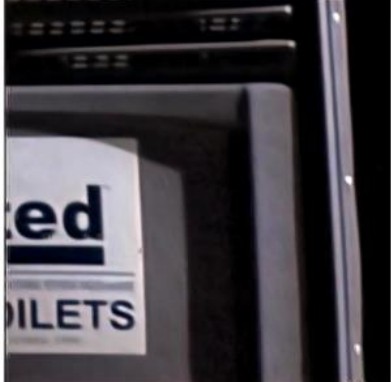

(d) HISP 1 Output
(DL + Sharpen)

(e) HISP 2 Output
(DL + Sharpen + Constrast)

(f) HISP 3 Output
(DL + Sharpen
+ Constrast + Denoise)

**Figure 8.** Ground truth image and output images of different pipelines.

The traditional submodules used to assist DL in image processing did improve the image quality, and the experimental results preliminarily confirmed the superiority of HISP over traditional ISP and DLISP, laying a theoretical foundation for subsequent implementation.

The ground truth image was used as the control group for this group of experiments. The results in black and bold in Table 2 are the best results in the experimental group, while the red ones are the worst.

**Table 2.** By using traditional ISP modules to assist in optimizing DLISP, the feasibility of HISP was preliminarily verified (the best-performing results are bold and in black in the table, and the worst-performing results are bold and in red).

|  | BRISQUE | PIQE | NIQE | NIMA | RankIQA | ABS | PSNR | SSIM |
|---|---|---|---|---|---|---|---|---|
| Gt | 42.62 | 69.36 | 5.03 | 4.79 | 2.97 | 8.70 | - | - |
| Traditional ISP Output | **25.56** | **24.76** | **3.35** | **3.43** | 2.66 | **5.30** | 10.0495 | **0.1087** |
| DLISP Output | 43.59 | 66.74 | 4.86 | 3.75 | **2.17** | 6.10 | 10.4020 | 0.3085 |
| DL Output + Sharpen | 44.44 | 76.30 | 8.03 | **4.14** | 2.45 | **7.90** | **10.4043** | **0.3171** |
| DL Output + Sharpen + Contrast | 43.08 | 77.30 | **8.20** | 4.03 | 2.72 | 6.40 | **9.5036** | 0.2178 |
| DL Output + Sharpen + Contrast +Denoise | **46.11** | **80.77** | 6.01 | 3.96 | **2.93** | 7.10 | 9.5323 | 0.2336 |

Due to the ability to deal with the problems of blurred edges and blurred information in the processing results of DLISP, the method of deep learning plus traditional sharpening submodules achieved the highest score in half of the total eight IQA methods, and it also had high scores under other IQA systems, giving relatively excellent performance. At the same time, the overall performance of the method of assisting deep learning processing through submodules was better than that of traditional ISP and DLISP, which also preliminarily proves the feasibility and excellence of HISP.

## 4. Implementation

While both AI algorithms and traditional ISP methods have their own advantages, we have also preliminarily verified the enormous potential of HISP, which may achieve a synergistic effect of 1 + 1 > 2. However, in current edge DLISP devices, the DL algorithm often needs to be run on an ARM core or even a PC workstation. A faster edge inference solution is urgently needed for HISP.

FPGAs have significant advantages in parallel computing [54], and accelerating algorithms through FPGAs can prevent the DL algorithm part of HISP from becoming a burden on speed, allowing the final HISP product to have sufficient real-time performance.

Implementing a general-purpose hardware accelerator like NVDLA ensures hardware versatility and generality, but it also requires software drivers and the acceleration ratio for a specific network may not be ideal. On the other hand, implementing a specific DPU for a neural network structure, starting from specific operators to implement a hardware network, emphasizes the specificity of the accelerator from the design stage and can achieve higher acceleration ratios. This approach also eliminates the need for running an operating system or compiling kernel drivers, resulting in high flexibility and a short development cycle [55].

Both two FPGA hardware solutions have been implemented simultaneously.

### 4.1. FPGA Implementation of NVDLA

On the AXU9EG development board, a minimal lightweight accelerator with an $8 \times 8$ MAC operation array was obtained by trimming the open-source RTL source code of NVDLA. Due to its complex logical core structure, a smaller scale needed to be trimmed to enable smooth driving on FPGA and reduce resource consumption.

Further optimization of the NVDLA RTL code was required. As NVDLA is aimed at ASIC design, the Verilog code for internal RAM is a structural level description, which means that instantiating RAM will consume a large amount of FPGA LUT resources. Therefore, all RAMs were replaced with FPGA internal block RAMs to reduce LUT overhead and improve operating speed.

A top-level file wrapper interface was then developed, and NVDLA was encapsulated as a callable custom IP core in Vivado. The 4-core ARM-Cortex-A53 CPU unit on the development board was interconnected with the NVDLA IP core through the CSB and AXI buses using block design.

The CSB was the control bus for NVDLA, and after the CSB2APB conversion module provided by NVIDIA converted the CSB protocol to the APB protocol, signals were extracted from NVDLA, and the AXI2APB Bridge IP core provided by Xilinx was used to convert the AXI-Master control lines extracted from the ARM CPU core into the APB protocol. This enabled the ARM core to control NVDLA through a memory-mapped mechanism to read and write NVDLA's registers.

Meanwhile, the ARM CPU core's AXI-Slaver was controlled by NVDLA's AXI-Master to enable NVDLA to access the DDR storage on the ARM CPU side, allowing shared memory and faster access speed (Figure 9).

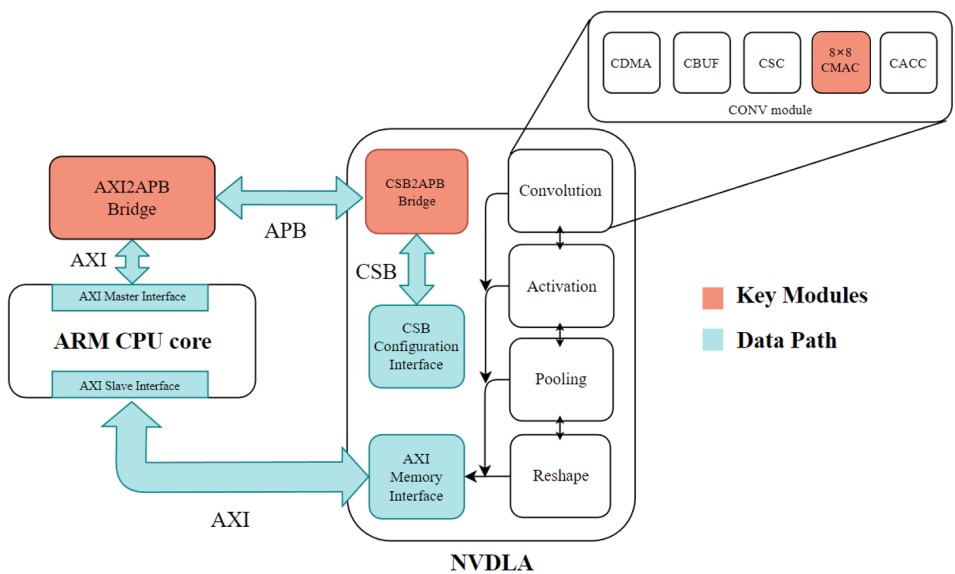

**Figure 9.** Implementation of NVDLA based on FPGA.

To run NVDLA, kernel driver and Runtime are necessary. We built a Linux operating system adapted for ARM cores using Petalinux, configured the device tree, added the compiled NVDLA kernel driver file, and replaced the file system with Ubuntu, which is more conducive to subsequent development.

Due to the limitation of hardware resources, it was not possible to implement a full version of NVDLA on an FPGA. Therefore, we configured and re-trimmed the accelerator core. Except for the use of the smallest $8 \times 8$ array, there was no SRAM interface, lookup table, and RUBIK engine enabled. Therefore, it did not support deconvolution, softmax, or even excessive pooling operations, and only supported one activation operator ReLU. In order to make up for the shortcomings of NVDLA's FPGA implementation in operator support, after successfully testing inference acceleration in Runtime, we chose Tengine, a framework developed by Open AI Lab, to deploy complex deep learning neural network models to the NVDLA hardware accelerator backend. Developed using C language under Tengine, it quickly and efficiently deployed models in formats including TensorFlow, PyTorch, and ONNX on various embedded devices. Additionally, it performed heterogeneous computing by utilizing both ARM CPUs and NVDLA through graph partitioning. As a result, Tengine schedules NVDLA and on-chip CPUs for heterogeneous computing, and in addition to basic convolution and activation, it also supports operators like deconvolution, concatenation, and pooling in end-to-end networks such as UNet. Figure 10 shows the Tengine's top-down technical architecture.

### 4.2. FPGA Implementation of a Dedicated DPU for UNet

To achieve dedicated acceleration of neural networks in FPGA, it is necessary to have an understanding of the network structure and split various operators into hardware structures. We chose the small and typical UNet neural network as an entry point and first

wrote corresponding operators for convolution, max pooling, and deconvolution using high-level synthesis (HLS) as shown in Figure 11.

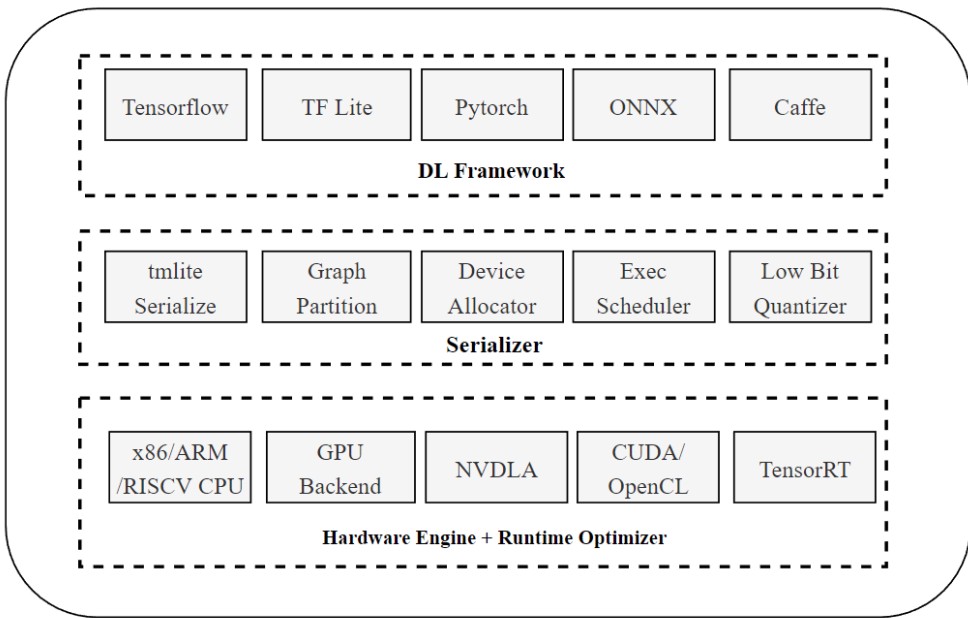

**Figure 10.** Tengine's top-down technical block diagram.

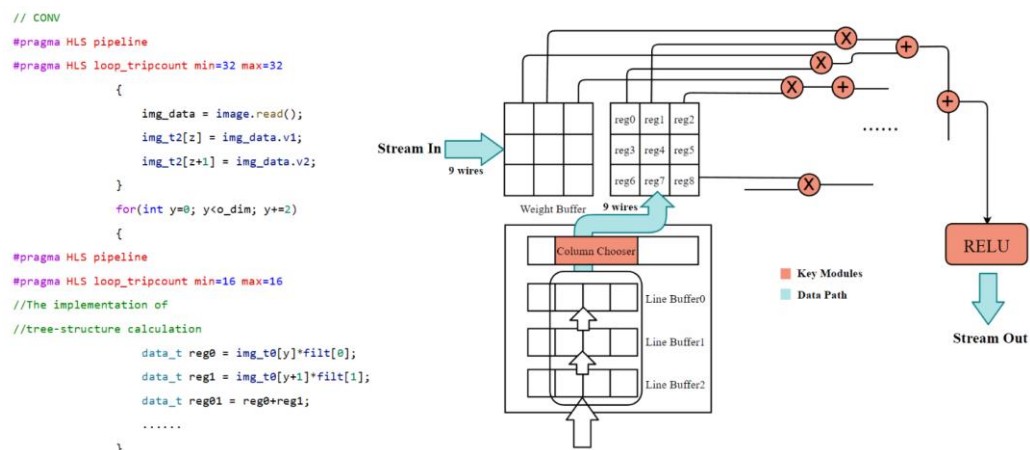

**Figure 11.** Key code and architecture of hardware convolution operator implemented through HLS. The multiplication and plus signs in the upper right corner represent the convolution process of the data stream.

The three operators compiled successfully in HLS were exported as custom IP cores and integrated into a block design in the Vivado IDE. The ARM core scheduled the three operators through the AXI bus in turn to achieve acceleration of the entire neural network inference. At the same time, DMA was set up to move data and ensure the flow of data, reducing the time required for memory access as shown in Figure 12.

### 4.3. FPGA Implementation of HISP Pipeline

Based on the specialized DPU, various traditional ISP submodules were implemented on FPGA.

In the previous section, we verified through comparative experiments that DL algorithms had better image restoration quality for Bayer domain images in low-light and

high-noise environments; the bad pixel correction (BPC), black level correction (BLC), lens shading correction (LSC), and Bayer noise reduction (BNR) were handled by DL algorithms. Therefore, the traditional ISP submodules used to assist DL algorithms only included an automatic white balance (AWB) module, denoising module, edge enhancement module (EE), and gamma correction module.

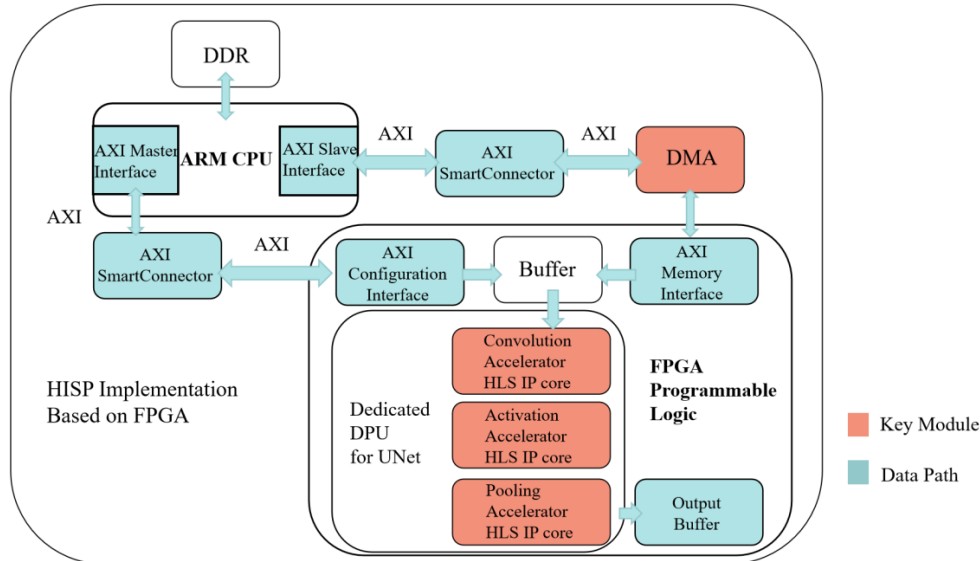

**Figure 12.** The architecture of UNet acceleration dedicated DPU which was implemented on FPGA.

There are many algorithms for automatic white balance (AWB), such as the gray world method, perfect reflector method, and dynamic threshold method. At present, there are many AWB studies through neural networks [56–58], but considering that the final implementation needs to be done through an FPGA, a simple calculation process for the algorithm was necessary. Among the aforementioned algorithms, the gray world method is not only simple to implement but also requires a small amount of computation, so it was ultimately chosen as the automatic white balance implementation method.

First, calculate the average values of the R, G, and B channels, and set the K value as half of the maximum value among the channels, which is 128. Then calculate the gains Kr, Kg, and Kb [59]:

$$
\begin{aligned}
K_r &= K/R_{aver}; \\
K_g &= K/G_{aver}; \\
K_b &= K/B_{aver};
\end{aligned}
\tag{6}
$$

The new pixel value is the sum of the gain and K:

$$
\begin{aligned}
R_{new} &= R \times K_r; \\
G_{new} &= G \times K_g; \\
B_{new} &= B \times K_b;
\end{aligned}
\tag{7}
$$

The denoise module uses a Gaussian filter to reduce noise, which follows a normal distribution (Gaussian white noise) introduced by the sensor. The Gaussian filter essentially performs a weighted average process on each pixel in the image. To implement this on an FPGA, we can first abstract this problem as a convolution process between the image and a Gaussian kernel [60]:

$$
I_\sigma = I \times G_\sigma
\tag{8}
$$

The Gaussian kernel is equal to:

$$
G_\sigma = \frac{1}{2\pi\sigma} e^{-(x^2+y^2)/2\sigma^2}
\tag{9}
$$

The edge enhancement module (EE) includes the Sobel operator, Laplacian operator, and Canny operator. The latest legacy algorithms also include FE [61], Edge Boxes [62], and SemiContour [63]. After comparing the results of the implementation, the Laplacian operator was ultimately chosen.

For a continuous function, the Laplacian operation is defined as:

$$\nabla^2 f = \frac{\partial^2 f}{\partial x^2} + \frac{\partial^2 f}{\partial y^2} \tag{10}$$

For digital images, the Laplacian operator can be simplified as:

$$g(i,j) = \sum_{r=-k}^{k} \sum_{s=-l}^{l} f(i-r, j-s) H(r,s), i,j = 0, 1, 2 \sim N-1 \tag{11}$$

The entire process can be seen as a convolution between the entire image and the Laplacian operator. When K = 1 and I = 1, H(r,s) represents the Laplacian operator with the following formula. H1 is the four-directional sharpening operator template, and H2 is the eight-partition template:

$$H1 = \begin{pmatrix} 0 & -1 & 0 \\ -1 & 4 & -1 \\ 0 & -1 & 0 \end{pmatrix} \quad H2 = \begin{pmatrix} -1 & -1 & -1 \\ -1 & 8 & -1 \\ -1 & -1 & -1 \end{pmatrix} \tag{12}$$

When implemented on an FPGA, the H2 operator needs to be stored in a 3 × 3 register group first. Then, the image is converted to grayscale, and the grayscale pixels are cached row by row into another 3 × 3 register group. The stored values of the corresponding position registers are multiplied by nine booth multipliers, and the results are added to the original pixels to obtain the new pixel values at the corresponding positions. The entire process is implemented through pipelining.

Gamma correction requires a nonlinear transformation of the brightness level to make the image's brightness and color more vivid. To implement gamma correction on an FPGA, a truth table needs to be stored in ROM to correspond to the gamma curve for lookup and to perform calculations for the nonlinear function.

After implementing all the submodules, they are connected to the hardware acceleration unit to form a system-level HISP design as shown in Figure 13.

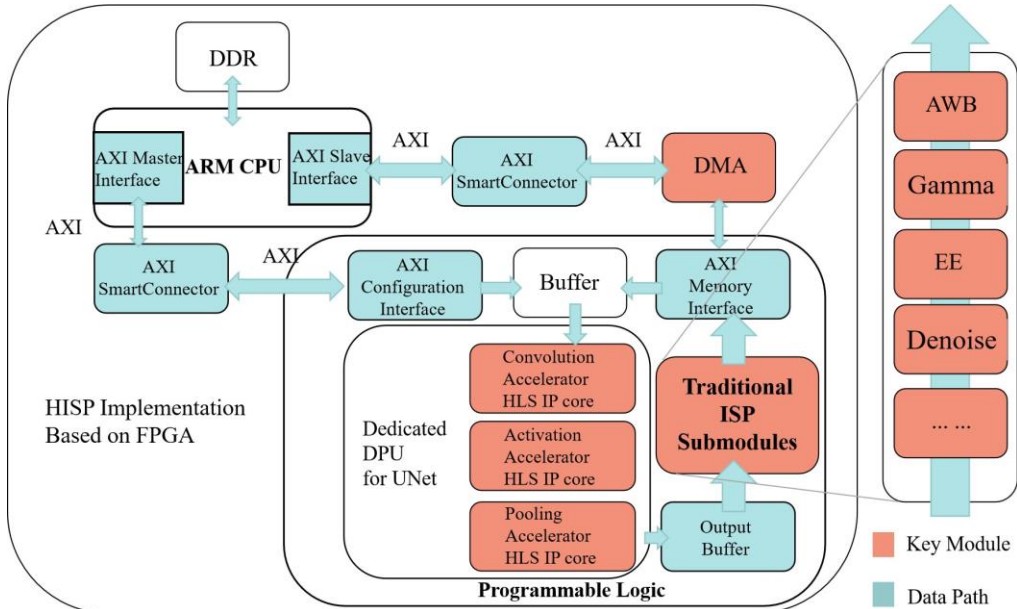

**Figure 13.** Implementation of HISP on FPGA.

## 5. Results

### 5.1. Optimal Acceleration Scheme

Comparing the three different approaches of using ARM CPU with Ubuntu OS and a Tengine framework, using ARM + Tengine to call DLA for acceleration, and using bare-metal scheduling of specialized DPU, we evaluated their performance from different perspectives.

We used different software frameworks to infer UNet under the Intel x86 i7 series CPU of a personal computer. The purpose was to use the inference speed of the 3600 MHz high-frequency CPU to give readers an idea of the scale and running time of the UNet network. Then, we compared it with the final DPU acceleration solution to reflect the superiority of the solution in terms of speed. At the same time, we also listed the running time of edge inference solutions such as ARM CPU and NVDLA, showing the dominant performance of the FPGA DPU implementation solution at the low frequency of 200 MHz at the edge.

For the edge inference scenario (ARM CPU and FPGA only), this study ran the same model of the same algorithm through ARM CPU, ARM CPU + NVDLA, and ARM CPU + DPU schemes. The dedicated DPU solution for UNet stood out with the lowest latency without any software framework support. As shown in Table 3, in the absence of accelerators, the edge ARM CPU took a full 3785.9 ms to calculate the large number of convolutions in the network. With the blessing of DPU, this number was reduced to 423.75 ms, achieving an acceleration ratio of 8.93×. Moreover, for deconvolution and max pool, the calculation latency of the DPU was as low as 2.46 ms and 97.10 ms, achieving an impressive speedup ratio of 46.30× and 36.49×, respectively. The total inference latency of the entire algorithm was reduced from 7675 ms to 523.28 ms, achieving a speedup ratio of 14.67×. The premise of the above results is that the ARM CPU inference is accelerated by parallel computing by the Tengine framework, and the DPU will only bring greater improvement to the edge CPU without deploying the deep learning framework.

**Table 3.** Latency and power consumption of the same UNet model running on different hardware, software, and operating systems (the best-performing results are bold in the table).

| Device | Frequency (MHz) | OS | Software | Latency (ms) | | | | Total On-Chip Power (Watt) |
|---|---|---|---|---|---|---|---|---|
| | | | | Conv | Maxpool | Deconv | Total | |
| x86 CPU | 3600 | Windows 10 | Python 3.6.4 | 127,835.99 | 2776.71 | 32,452.11 | 166,625.00 | 125 |
| x86 CPU | 3600 | Windows 10 | C (gcc 8.1) | 16,679.21 | 11.47 | 3829.87 | 21,551.00 | 125 |
| x86 CPU | 3600 | Ubuntu 18.04 | Tengine Lite 1.0 | **289.00** | 5.40 | 287.30 | 609.36 | 125 |
| ARM CPU | 1333 | Ubuntu 18.04 | Tengine Lite 1.0 | 3785.90 | 113.9 | 3543.50 | 7675.00 | **0.52** |
| ARM CPU + DLA on FPGA | 1333 & 200 | Ubuntu 18.04 | Tengine Lite 1.0 | 2958.32 | 79.73 | 2763.39 | 6007.23 | 3.85 |
| ARM CPU + DPU on FPGA | 1333 & 200 | - | - | 423.75 | **2.46** | **97.10** | **523.28** | 4.04 |

The dedicated DPU solution was not only faster than the results of ARM or even Intel i7 x86 CPUs but also had the best performance in terms of power consumption, development cycle, and flexibility, making it undoubtedly the best choice for accelerating deep learning algorithms in HISP.

### 5.2. Optimal Task Allocation Scheme

Due to the fact that the DPU mainly served the purpose of improving the algorithm inference speed and increasing the lower limit of HISP's real-time performance, the main metric we focused on in designing the DPU was the inference speed. However, when

designing the entire HISP pipeline, we needed to consider the image processing effects, latency, and hardware resource consumption of the added modules as a whole. The intuitive output is shown in Figure 14.

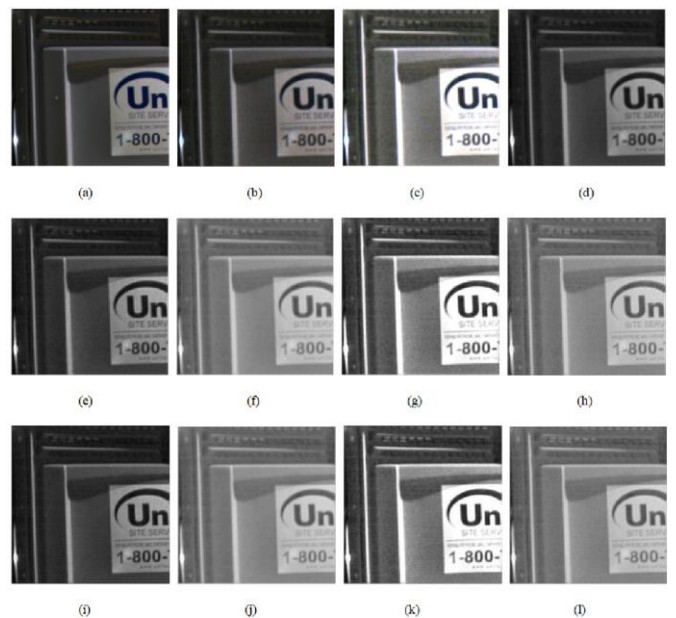

(a) Ground Truth
(b) Output by DLISP
(c) HISP Pipeline 1: DL+AWB
(d) HISP Pipeline 2: DL+Denoise
(e) HISP Pipeline 3: DL+EE
(f) HISP Pipeline 4: DL+Gamma
(g) HISP Pipeline 5: DL+AWB+EE
(h) HISP Pipeline 6: DL+Gamma+EE
(i) HISP Pipeline 7: DL+EE+Denoise
(j) HISP Pipeline 8: DL+Gamma+Denoise
(k) HISP Pipeline 9: DL+AWB+EE+Denoise
(l) HISP Pipeline 10: DL+Gamma+EE+Denoise

**Figure 14.** Output images of 10 different HISP implementations.

In order to facilitate the experiment, reduce the time of simulation and iteration, and at the same time not affect the effect of UNet when processing images, we chose the input resolution of $256 \times 256$ and specially designed it. In actual application scenarios, latency, power consumption, and resource consumption may increase. Table 4 Compares the outputs of different pipelines using a no-reference IQA scoring system.

**Table 4.** Comparing the ground truth, DLISP output, and the output of 10 HISP pipelines using a no-reference IQA scoring system (the best-performing results are bold in the table).

|  | BRISQUE | PIQE | NIQE | NIMA | RankIQA | ABS |
|---|---|---|---|---|---|---|
| Gt | 42.62 | 69.36 | 5.03 | 4.79 | 2.97 | 8.70 |
| DL | 46.44 | 80.92 | 5.52 | 3.13 | 2.19 | 5.40 |
| Pipeline 1 | 51.01 | 77.08 | 5.53 | 3.44 | **2.50** | 7.90 |
| Pipeline 2 | 54.93 | 83.12 | 5.00 | **3.50** | 2.04 | 6.50 |
| Pipeline 3 | 45.01 | 60.21 | 5.42 | 3.59 | 1.92 | 6.20 |
| Pipeline 4 | 49.82 | 78.45 | 5.58 | 3.20 | 1.94 | 3.10 |
| Pipeline 5 | **39.90** | **52.73** | 5.70 | 3.47 | 2.31 | **8.30** |
| Pipeline 6 | 45.57 | 57.26 | 5.15 | 3.41 | 1.90 | 4.70 |
| Pipeline 7 | 50.30 | 69.20 | 4.93 | 3.27 | 2.18 | 7.00 |
| Pipeline 8 | 57.01 | 80.89 | 4.99 | 3.23 | 1.81 | 4.00 |
| Pipeline 9 | 46.11 | 68.27 | **4.79** | 3.25 | 2.38 | 7.70 |
| Pipeline 10 | 54.70 | 66.09 | 4.90 | 3.19 | 1.92 | 4.60 |

In the whole HISP, the DPU realizes the real low power consumption. In the end, it is part of the traditional module that really determines the final power consumption, and the final power consumption and resources will double with the increase of the input image resolution.

Therefore, when implementing the traditional module, we focused on the speed of inference, the consumption of hardware resources, and the size of the power consumption as shown in Table 5.

**Table 5.** On-chip hardware resource consumption and power consumption of traditional modules in the 10 HISP pipelines.

| | Lantency (Microseconds) | Hardware Resource Consumption | | | Power (Watts) |
|---|---|---|---|---|---|
| | | **LUTs** | **Registers** | **BRAMs** | |
| Pipeline 1 | 968 | 120 | 182 | 0 | 4.611 |
| Pipeline 2 | 965 | 128 | 216 | 0 | 6.319 |
| Pipeline 3 | 970 | 327 | 319 | 0 | 6.964 |
| Pipeline 4 | 968 | 213 | 52 | 1.5 | 3.823 |
| Pipeline 5 | 1650 | 376 | 423 | 0 | 8.562 |
| Pipeline 6 | 1651 | 492 | 293 | 1.5 | 7.712 |
| Pipeline 7 | 1657 | 384 | 457 | 0 | 8.575 |
| Pipeline 8 | 1655 | 270 | 190 | 1.5 | 5.435 |
| Pipeline 9 | 2327 | 433 | 527 | 0 | 10.182 |
| Pipeline 10 | 2328 | 526 | 561 | 1.5 | 9.313 |

In this part, several typical RGB domain image processing submodules including AWB, gamma, EE, and denoise were connected with the DPU part to form a total of 10 different HISP pipelines.

First of all, in order to give a definite solution to the optimal division of labor of HISP, it was necessary to perform IQA on the output of each HISP pipeline. Through the previous analysis and demonstration, it was preliminarily found that the edge enhancement (EE, i.e., sharpening) module can significantly improve the quality of the image output by the neural network model. The results proved that the effect of edge enhancement (EE) was not disappointing. Among the 10 implementation schemes, the average score of the result of adding the EE unit was 0.02 higher than that of the experimental group without it in the NIMA IQA system and 1.64 higher in the ABS evaluation system.

For brightness and color processing, we selected two modules, AWB and gamma. The experiment showed that under these six types of IQA systems, the average score of the pipeline participated by the AWB module was 0.13, 0.56, and 3.87 higher than that of gamma in NIMA, RankIQA, and ABS.

In RankIQA, the simple solution of DPU + AWB was at the top of the list. At the same time, in the manual blind evaluation, it was obvious that AWB greatly improved the amount of image information that the human eye can perceive, which is far higher than the impact achieved by other modules.

It can be said that in the face of image processing with low light and high noise, AWB is an absolutely indispensable module of HISP as shown in Figure 15.

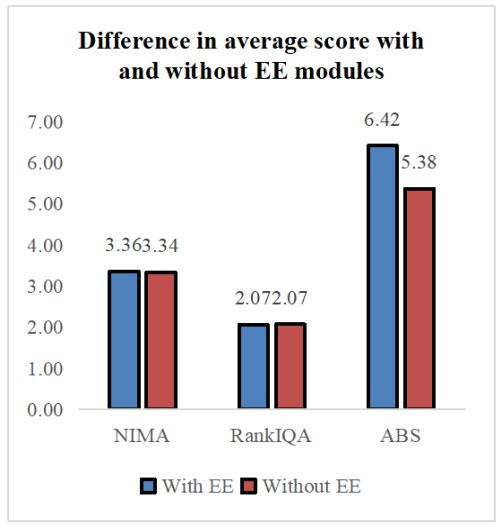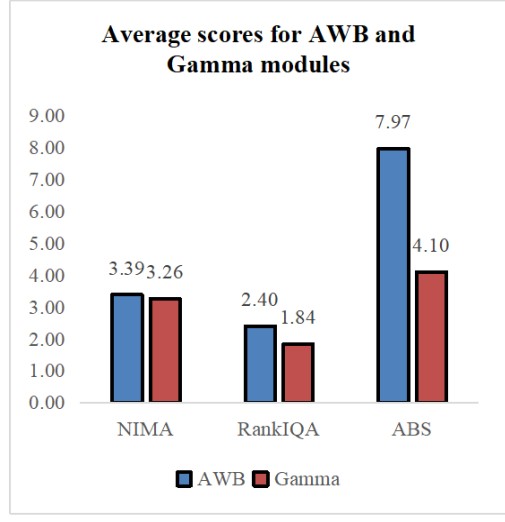

**Figure 15.** The specific performance of the EE, AWB, and gamma modules in HISP.

At the same time, the denoise module did not play the key role as imagined. In each pipeline, the addition of the denoise module showed little improvement in image quality. The specific improvement is shown in Figure 16.

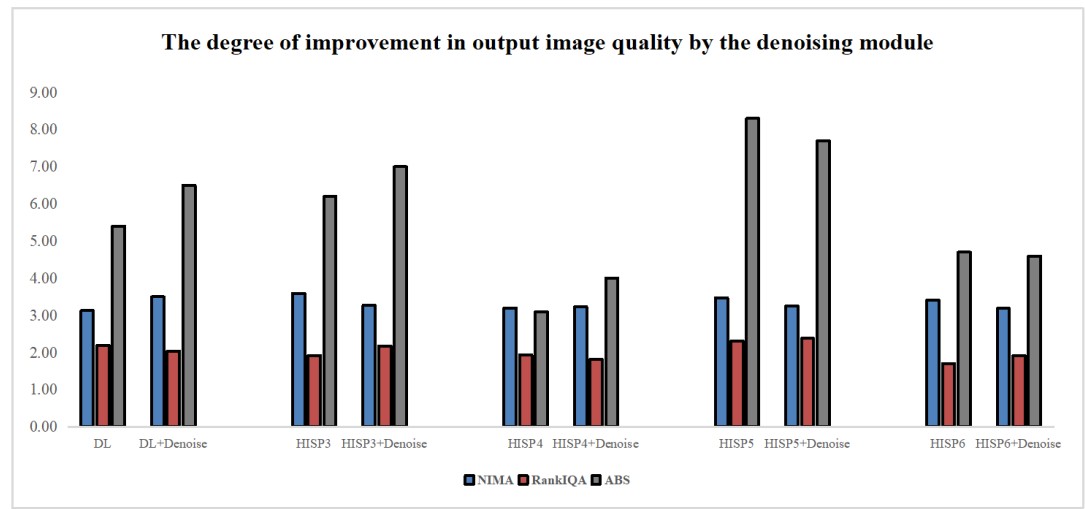

**Figure 16.** The impact of adding a denoising module to the image quality in HISP. Despite consuming additional resources, the score improvement was minimal or even decreased.

Among the 10 HISP pipelines, as shown in Figure 17, the implementation of HISP pipeline5: DPU + AWB + EE scheme achieved the best performance and scored 39.90, 52.73, and 8.3 points, respectively, in the three IQA systems, BRISQUE, PIQE, and ABS. Its performance in BRISQUE and PIQE even exceeded that of the ground truth image, and the score in NIMA was also close to the highest score.

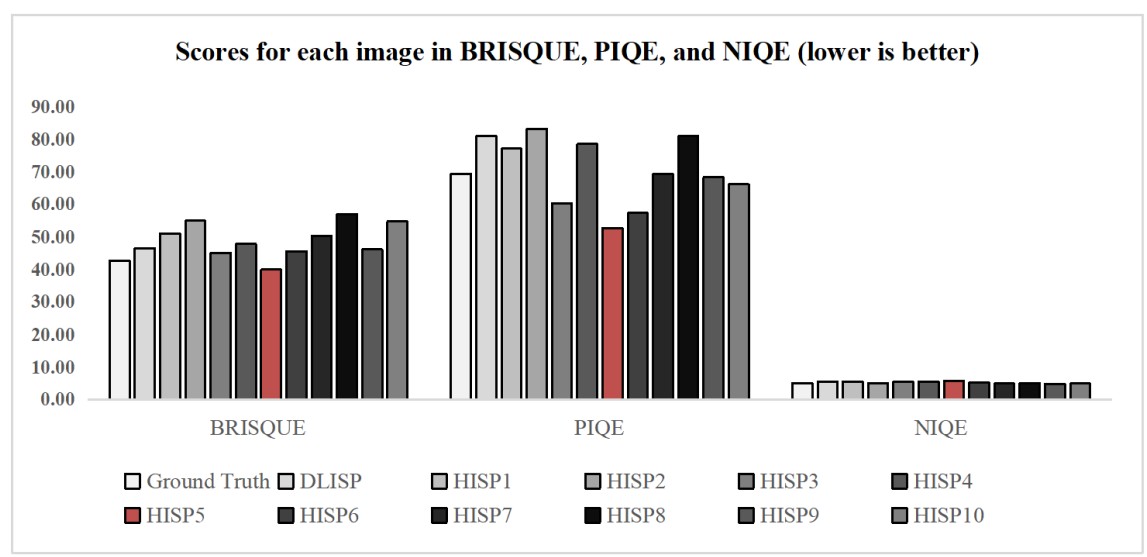

**Figure 17.** HISP pipeline5: DPU + AWB + EE shows excellent performance in traditional IQA.

In addition to processing effects and output quality, we also needed to pay attention to the resource consumption and time delay of hardware implementation. As the size of the image increased, the consumption of various resources and power on the FPGA increased exponentially, so these must also be taken into account if you want to obtain the optimal effect when implementing it. For instance, the gamma module and AWB module had similar power consumption and resource consumption, and AWB had a significantly better effect, while denoise did not play an ideal role in the case of consuming a lot of resources and delay.

Therefore, in terms of the quality of the output image alone, AWB and EE were the modules with the best effect and the greatest impact, while modules such as gamma and denoise did not bring considerable improvement while consuming additional resources.

Considering the optimal effect plus reasonable power consumption and resource consumption, the optimal division of labor for HISP should be to hand over bad pixel correction (BPC), black level correction (BLC), lens shading correction (LSC), and Bayer noise removal (BNR) and demosaicing functions to the deep learning algorithm for reasoning, and on this basis let the AWB and EE modules be responsible for image quality in the RGB domain's further improvement.

The optimal structure of HISP should be DPU + AWB + EE. Under this structure, implementation through FPGA consumed 376 LUTs and 423 register resources, the entire HISP pipeline delay was 524.93 ms, and achieved a BRISQUE score of 39.90, PIQE score of 52.73, NIQE score of 5.70, RankIQA score of 3.47, NIMA score of 2.31, and ABS score of 8.3.

In the HISP implemented by FPGA, the inference time of the DPU part was 523.28 ms, and the power consumption was 4.04 watts. The pushing time of the traditional module pipeline was 1.65 ms, and the power consumption was 8.56 watts. The total inference time of HISP was 524.93 ms, and the total power consumption was 12.6 watts. For edge inference, HISP provides a reliable and efficient solution.

## 6. Conclusions and Future Work

### 6.1. Conclusions

Based on the challenges of edge image processing, this paper proposes a novel heterogeneous image signal processor (HISP) pipeline that combines traditional ISP and deep learning ISP (DLISP) techniques. The proposed pipeline provides a low-cost and fully replicable solution for edge image processing, achieving a BRISQUE score of 39.90, PIQE score of 52.73, NIQE score of 5.70, RankIQA score of 3.47, NIMA score of 2.31, ABS score of 8.3, and a single processing time of 524.93 ms with only 8.56 W power.

The paper has three main contributions. First, it provides a detailed analysis of the strengths and weaknesses of traditional ISP and DLISP, and proposes the concept of HISP to combine the two, leveraging their advantages while minimizing their drawbacks. Second, the paper integrates different traditional ISP modules with DLISP to create multiple pipelines, which are evaluated through multiple dimensions of image quality assessment (IQA). The paper proposes the HISP allocation plan that achieves the optimal balance among processing speed, resource consumption, and development difficulty. Third, the paper implements a dedicated DPU for UNet on FPGA, achieving a $14.67\times$ acceleration ratio. Additionally, the paper details design of a heterogeneous ISP that combines traditional ISP and DLISP based on the optimal division of labor, all on FPGA, resulting in the best image quality in edge scenarios.

The research shows that the proposed HISP pipeline is effective in edge image processing scenarios and can be replicated as low-cost solutions. The combination of traditional ISP and DLISP not only minimized their drawbacks but also improved the overall performance of image processing. The use of FPGA and specific DPU for UNet also greatly improved the efficiency of deep learning processing in edge scenarios.

In conclusion, this paper presents a promising solution for edge image processing, combining traditional ISP and DLISP techniques in a heterogeneous image signal processor pipeline. This research provides important insights into the challenges and opportunities of edge image processing, and offers a roadmap for the development of low-cost and effective solutions for edge image processing.

### 6.2. Future Work

The current tests and experiments are mainly faced with extreme scenes of low light and high noise. Although these two types of scenes show the advantages of DLISP and HISP in image processing, to completely replace the traditional method, it is still necessary to conduct extensive data collection and comparative experiments on common scenes

to ensure the versatility of the final product. Moreover, deep learning algorithms are evolving at an incredible pace. Applications that require cameras such as autonomous driving, metaverse, drones, and VR are also opening up a broad market for ISP algorithms. Because algorithm applications such as YOLOv5-tassel and HYDRO-3D continue to emerge, HISP will face more challenges in different application scenarios. According to the latest research, we are focusing on new directions and next-generation technologies, including quantum artificial intelligence, which typically leads to astonishing nonlinear classification capabilities, robustness to noise, and better signal processing results [64–67]. Therefore, designing more universal and novel algorithms is an important task in the future.

In addition, the IQA scheme we established mainly adopts the existing general scheme. The texture detail of an image with a high PSNR or SSIM score does not necessarily correspond to the visual habits of the human eye. Therefore, more effective image quality indicators are synthesized in this study. However, the above problems still exist. For example, according to interviews with reviewers in the ABS scoring system, we found that EE provides an extremely obvious and intuitive improvement in the perception of human vision, even if humans do not know what has changed in the image, but at first glance, they can feel that the picture passing through EE is clearer and more detailed. Furthermore, it can be seen from the experiments that the evaluation results of various latitudes are often inconsistent, and the quantified scores are not accurate enough. Therefore, we hope to formulate a follow-up image quality evaluation system in edge scenes, through the combination of traditional algorithms and deep learning methods, that can obtain the results that are most consistent with naked-eye vision and CV algorithms.

Furthermore, the results of this project are applicable to edge inference, which is of great significance to edge image processing. However, the implementation on FPGA requires sufficient experience and workload. In the future, we plan to develop a toolchain that will accelerate the end-to-end implementation of algorithms to FPGA HISP and facilitate short-cycle product development.

Finally, we will try our best to cooperate with enterprises to realize tape-out and mass production of HISP products on the basis of complete verification.

**Author Contributions:** Conceptualization, J.C. and W.L.; methodology, J.C. and W.L.; software, J.C., B.W. and X.S.; validation, J.C., B.W. and S.H.; formal analysis, J.C., B.W. and S.H.; investigation, J.C., B.W. and W.L.; resources, W.L. and G.G.; data curation, J.C. and B.W.; writing—original draft preparation, J.C., B.W. and Q.X.; writing—review and editing, J.C., B.W., W.L., G.G., S.H. and X.S.; visualization, J.C., S.H. and Q.X.; supervision, S.H, Q.X., G.G. and W.L.; project administration, J.C. and W.L.; funding acquisition, W.L. and G.G. All authors have read and agreed to the published version of the manuscript.

**Funding:** This work was supported by the National Key R & D Program (Grant No. 2022YFA1402503), the Special Fund of Hubei Luojia Laboratory (Grant No. 220100025), the Key Project of Hubei Province (Grant No. 2021BAA179), the Fundamental Research Funds for the Central Universities (Grant No. 413000137) and Hubei Province Technology Innovation Project (Grant No. 2022BEC035).

**Data Availability Statement:** The data that support the findings of this study are available from the author J.C., upon reasonable request.

**Conflicts of Interest:** The authors declare no conflict of interest.

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
