# Peer review of "HISP: Heterogeneous Image Signal Processor Pipeline Combining Traditional and Deep Learning Algorithms Implemented on FPGA"

_electronics, doi:10.3390/electronics12163525_

Round 1

Reviewer 1 Report

The manuscript “HISP: Heterogeneous Image Signal Processor Pipeline Combining Traditional and Deep Learning Algorithms Implemented on FPGA” proposes a novel Heterogeneous Image Signal Processor (HISP) pipeline that combines the advantages of traditional image signal processor and Deep Learning ISP (DLISP). The author elucidates the potential and applicability of the proposed HISP approach in the realm of edge image processing. Such an approach offers a roadmap for the development of cost-effective and efficient solutions in this domain. However, there are some elements that are still unclear and the manuscript lacks some significant research contents. In addition, my suggestions are as follows (see list below).

1. When introducing related work about deep learning ISP, the author should add necessary citation accordingly.

2. In Figure 15, when employing the NIMA and RankIQA assessment systems, the influence introduced by the EE module is nearly negligible. However, when utilizing the ABS evaluation system, the impact of the EE module becomes more pronounced. The rationale behind this phenomenon warrants further investigation.

3. In the manuscript, the abbreviation of “bad pixel correction” is DPC. Is such abbreviation right? If not, please correct it.

4. In “implementation” section, the author state that "NVDLA has limited operator support on FPGAs." It is recommended to provide more context about implications of these limitations for your implementation.

5. The cross combination of artificial intelligence and quantum physics (Quantum artificial intelligence) often leads to better results in image processing. Mention that quantum artificial intelligence is a promising direction. Refer to recent interesting works [r1-r4] in this area. The authors can make some discussion.

[r1] Quantum Deep Learning Neural Networks, Advances in Information and Communication 2, 299-311 (2020).

[r2] Experimental quantum advantage with quantum coupon collector, Research 2022, 9798679 (2022).

[r3] Quantum machine learning beyond kernel methods, Nature Commun.14, 517 (2023).

[r4] Quantum Neural Network for Quantum Neural Computing, Research 6, 0134 (2023).

6. It is noted that the abstract and conclusion emphasize a significant contribution - achieving a 14.67x acceleration ratio by implementing DPU on FPGA. However, in the 'Results' section, this acceleration is not explicitly mentioned. I recommend the authors to enhance the 'Results' section by including supplementary discussion and emphasizing the relevant data to substantiate this contribution.

7. The manuscript requires significant improvements in terms of language and style. There are several instances of unclear or convoluted sentences, grammatical errors, and inconsistent terminology.

For instance, in the abstract, "quanlity" should be corrected to "quality," and there is a missing essential space between "Field Programmed Gate Array" and "(FPGA)."

 Moderate editing of English language required

Reviewer 2 Report

This paper proposes a novel ISP pipeline that combines deep learning with traditional methods. Some modifications should be made before publication:

1) In the introduction, the application of camera image signal processing should be optimized, such as in the area of robotics and intelligent transportation: an automated driving systems data acquisition and analytics platform, integrated inertial-lidar-based map matching localization for varying environments, yolov5-tassel: detecting tassels in rgb uav imagery with improved yolov5 based on transfer learning, hydro-3d: hybrid object detection and tracking for cooperative perception using 3d lidar. Thus, these applications should be included in the introduction.

2) The writing style in the introduction needs improvement. The entire content is divided into too many small paragraphs.

3) Some of the content requires essential references, such as formulas (1)-(4), which are traditional formulas in the field of image processing. 

4) For Figure 17, you can choose colors with higher color contrast to visualize different results.

5) Is the metric based on Formula 5 your own design or based on previous work? If it is based on previous work, please kindly provide the corresponding reference. If it is your own design, please provide the reasoning behind your design.

6) The clarity in Figure 5 needs improvement, especially regarding the consistency of the fonts used in the figure.

7) Overall, this article's work is solid. It simultaneously considers algorithm implementation and hardware experiments. This work will have a positive impact on the development of edge image processing.

Round 2

Reviewer 1 Report

The authors have solved my issues. I recommend this manuscript for publication.  

Besides, I suggest that the authors pay attention to proofreading the volume number and page number of the reference..

  •  
  •  

Minor editing of English language required

Reviewer 3 Report

The paper presents the modification required.